# Shared genetic factors between osteoarthritis and cardiovascular disease may underlie common etiology

Karin Magnusson [1,2] ✉, Aleksandra Turkiewicz [1], Andrea Dell'Isola[1] & Martin Englund [1]

Osteoarthritis is one of the most common musculoskeletal diseases and increases the risk of severe cardiovascular disease, like heart attack and stroke. In some individuals, osteoarthritis and cardiovascular disease will co-occur. This co-occurrence might be due to shared risk factors, for example high age, lifestyle factors and/or a shared genetic liability for the two diseases. Here, we show that the correlation between osteoarthritis and cardiovascular disease can be explained by shared genetic factors, independent of high age and body weight, and also likely independent of lifestyle factors, like smoking and physical activity level. Findings suggest that genetic factors that are shared for osteoarthritis and cardiovascular disease may contribute to both diseases. Thus, the prevailing idea that osteoarthritis is predominantly a risk factor for cardiovascular disease is challenged. Our findings imply that the current diagnostic boundaries between these diseases may need to be re-evaluated.

Musculoskeletal conditions are among the most common causes of disability worldwide, with osteoarthritis (OA) being one of the most prevalent. OA is associated with a range of comorbidities, but cardiovascular disease (CVD) is considered to be one of the most important consequences of OA. In a meta-analysis of observational studies, the risk of CVD was increased by 12% to 37% in patients with OA compared with the general population[1]. Nonetheless, reverse associations cannot be ruled out[2,3]. It is currently unknown to what extent they share the same etiology in terms of genetic liability.

Family-based studies are useful in disentangling the relative contribution of genetic and environmental factors to a disease as they allow for the estimation of heritability, i.e. the total genetic contribution to a trait[4]. Several twin studies have revealed a potential high genetic contribution to OA, ranging from 45% up to 80%[5–9]. However, environmental factors like body composition, obesity and joint injury are also at play[6,9]. Similar etiological factors—i.e. a combination of genetics and environmental factors—are believed to be of relevance also in the etiology of the most common severe CVDs, like coronary heart disease, stroke, and heart failure. In studies of >20,000 Swedish twins, 50% of the variation in coronary heart disease mortality was reported to be explained by genetics and 50% by environmental factors[10,11]. The most important known environmental risk factors for CVD include obesity, tobacco use and physical inactivity[12], which are also risk factors for OA.

In some individuals, OA and severe CVD will co-occur. This co-occurrence might be due to shared risk factors, for example high age and high body weight, shared genetic liability, or both. Using twin studies, our aim was to examine to what extent the two conditions co-occur, and whether this co-occurrence is due to shared genetics vs due to environmental factors. Here, we show that this co-occurrence between osteoarthritis and cardiovascular disease can be explained by genetic factors that are shared for the two diseases, independent of high age and body weight, and also likely independent of lifestyle factors, like smoking and physical activity level.

## Results

Out of the available 80 740 twins aged 18 years or more in 1997, we studied 59,970 twins in 29,985 twin pairs with both twins alive at start

[1]Lund University, Faculty of Medicine, Department of Clinical Sciences Lund, Orthopaedics, Clinical Epidemiology Unit, Lund, Remissgatan 4, 222 42 Lund, Sweden. [2]Norwegian Institute of Public Health, Cluster for Health Services Research, Sandakerveien 24C, 0473 Oslo, Norway. ✉e-mail: karin.magnusson@med.lu.se

of follow-up and alive for at least one year after start of follow-up (main analyses). Thus, we excluded 10,810 singletons and 9,960 twins in pairs in which one or both twins were dead at the start of follow-up or during the first year. In total 7363 individuals had OA in any joint (of which 938, 2498 and 3845 in the hand, hip and knee joints, respectively, and as previously described in site-specific analyses[6,13,14]). Further, 13,685 individuals had any of the severe CVDs (of which 4855 had cardiac arrythmias, 6560 had CHD, 2724 had heart failure and 4730 had stroke). The mean (SD) age at start of follow-up was 48[15] years and somewhat varying across twins who were concordant for the different diagnoses (Table 1). BMI was available for 57,333 (96%) of individuals and when studied at pair level, for 55,144 twins (92%). Twins with missing BMI data were more often males (54%) and were on average younger (mean [SD] age 40.4 [19.5]) than observed in the full sample as described in Table 1. Pairs where at least one twin had missing BMI data were excluded from analyses that included BMI.

Mean BMI was generally lower in the total sample than among pairs concordant for OA and CVD (Table 1). Further, 2% vs 30% of individuals had missing data on smoking and physical activity level, respectively (4% vs. 36% on pair level). Because of the dependency in twin data and complexity in analyses, we avoided data imputation and performed complete case analysis only. The available data may imply some differences in lifestyle across combinations of diagnoses (Table 1).

OA and CHD were the most prevalent combined diagnoses (Table 1). For both OA and each of the CVDs, phenotypic correlations were generally low. Concordance for the diagnoses were generally higher in MZ twins than in DZ twins (Table 2), also across sex (S-Table 3). In total 13,074 (21.8%) died after start of follow-up. Of these deaths, 3599 (27%) were due to any of the included CVDs.

### Any site OA and any of severe CVD

The crude phenotypic correlation (rPT) between any site OA and any of the severe CVDs was 0.16 (95% CI = 0.14–0.18) indicating that there are common etiological influences (Table 3, Fig. 1). The MZ cross-twin-cross-trait correlation (rMZ=0.16 [95% CI = 0.13–0.19]) was equally high as the phenotypic correlation, yet higher than the DZ cross-twin-cross-trait correlation (rDZ=0.11 [95% CI = 0.08–0.13]) (Table 3). This pattern suggests that genetic factors are of importance in the shared etiology

of the two diseases.

High age (mean [SD] 61 [9.5] years) in individuals/pairs with both OA and severe CVD (Table 1) explained the lower phenotypic correlation in adjusted vs in crude models, whereas BMI was of less relevance (Fig. 1). The age- and sex-adjusted vs age-, sex- and BMI-adjusted phenotypic correlations between any site OA and any of CVDs were lower than the crude, at 0.09 (95% CI = 0.08–0.11) and 0.07 (95% CI = 0.04–0.10), respectively, with a similar pattern of rPT vs rMZ and rDZ ratios as in the crude analyses (Table 3, Fig. 1). Of the age- and sex-adjusted phenotypic correlations, 85% (95% CI = 55%–122%) (rPT=0.0802 [95% CI = 0.05–0.11]) could be explained by additive genetic factors, 0% (95% CI = −11%–11%) (rPT=0.0 [95% CI = −0.01–0.01]) by shared environmental factors and 15% (95% CI = −15%–42%) (rPT=0.014 [95% CI = −0.014–0.04]) by unique environmental factors (Fig. 1). In age-, sex- and BMI-adjusted analyses, additive genetic factors explained up to 100% (95% CI = 57%–143%) (Fig. 1), as calculated from [√A1 * Ra * √A2)/rPT] (S-Fig. 1): heritability for OA (√0.475) = 0.689 * genetic correlation between OA and CVD = 0.178 * heritability for CVD = (√0.328) = 0.572))/total phenotypic correlation=0.07. Measures of model fit are presented in S-Table 4.

We found no indication of sex differences in the shared and unique factors influencing any-site OA and any of the severe CVDs (S-Table 5, S-Table 6). Opposite-sex DZ twins had a similar cross-trait cross-twin correlation as same-sex DZ twins, indicating no qualitative sex-differences (S-Table 5). Co-occurrent any-site OA and any of the severe CVDs was more common in males than in females (higher rPT in males as shown in S-Table 6). For both sexes, the MZ cross-twin-cross-trait correlation was higher than the DZ cross-twin-cross-trait correlation, suggesting a common genetic contribution to OA and severe CVD in both sexes (S-Table 6).

In sensitivity analyses of pairs in which both twins survived through the entire follow-up, we included 7017 MZ twin pairs and 13,924 DZ twin pairs, i.e. 41,882 individuals (mean (SD) age 41.3 (13.0), 54% females). The numbers of twins concordant for both OA and each of the severe CVDs were lower than in the main analyses, yet again, generally higher for MZ pairs than for DZ pairs (S-Table 7). In this sample, we found a similar phenotypic correlation (age- and sex-adjusted rPT=0.07 [95% CI = 0.04–0.09]), again with indications of a

**Table 1 | Descriptive characteristics, individual level**

| | All | Members of pairs concordant for two traits | | | | | | |
| | | Any-site OA and any of CVDs | Any-site OA and card. Arr. | Any-site OA and CHD | Any-site OA and heart failure | Any-site OA and stroke | Hip OA and any of CVDs | Knee OA and any of CVDs |
|---|---|---|---|---|---|---|---|---|
| Nr of pairs | N = 29985 | N = 145 | N = 39 | N = 49 | N = 15 | N = 18 | N = 38 | N = 44 |
| Nr of individuals | N = 59 970 | N = 290 | N = 78 | N = 98 | N = 30 | N = 36 | N = 76 | N = 88 |
| Age at start of follow-up, mean (SD) | 48 (16) | 61 (10) | 59 (8) | 59 (9) | 67 (7) | 68 (8) | 63 (9) | 61 (9) |
| Sex, n | 32421 | 140 | 38 | 37 | 17 | 20 | 43 | 35 |
| % | 54 | 48 | 49 | 38 | 57 | 56 | 57 | 40 |
| College/university (>1 years), n | 16963 | 56 | 22 | 23 | 4 | 6 | 15 | 15 |
| % | 28 | 19 | 28 | 23 | 13 | 17 | 20 | 17 |
| BMI, mean (SD)* | 22.8 (5.9) | 25.3 (4.5) | 25.4 (4.4) | 25.8 (4.2) | 26.1 (4.5) | 23.4 (6.4) | 24.7 (4.6) | 26.6 (3.2) |
| Overweight, ≥25 kg/m2, n | 17854 | 149 | 39 | 58 | 17 | 16 | 33 | 60 |
| % | 30 | 51 | 50 | 59 | 57 | 44 | 43 | 68 |
| Current/former regular smoker, n | 33380 | 154 | 45 | 51 | 23 | 15 | 39 | 44 |
| % | 57 | 53 | 58 | 52 | 77 | 42 | 51 | 50 |
| Physically inactive, n | 25757 | 129 | 38 | 46 | 16 | 16 | 16 | 36 |
| % | 62 | 62 | 60 | 67 | 64 | 62 | 65 | 53 |

*OA* osteoarthritis, *CVD* cardiovascular disease, *CHD* coronary heart disease, *SD* standard deviation, *MZ* monozygotic, *DZ* dizygotic. *BMI data is missing for 8.05% of study population, smoking data is missing for 2.12% of individuals, physical activity data is missing for 30.5% of individuals.

**Table 2 | Number (percentage of all MZ/DZ twins, respectively) of pairs concordant for both osteoarthritis (OA) and cardiovascular disease (CVD)**

|  | Any site OA | | Hip OA | | Knee OA | |
|---|---|---|---|---|---|---|
|  | MZ conc. pairs | DZ conc. pairs | MZ conc. pairs | DZ conc. pairs | MZ conc. pairs | DZ conc. pairs |
| Any of CVDs | 64 (0.68) | 81 (0.39) | 18 (0.19) | 20 (0.10) | 22 (0.23) | 22 (0.11) |
| Card. arr. | 25 (0.26) | 14 (0.07) | 7 (0.07) | 2 (0.01) | 9 (0.09) | 5 (0.02) |
| CHD | 23 (0.24) | 26 (0.13) | 6 (0.06) | 4 (0.02) | 5 (0.05) | 9 (0.04) |
| Heart failure | 11 (0.12) | 4 (0.02) | 7 (0.07) | 0 (0.0) | 3 (0.03) | 0 (0.0) |
| Stroke | 6 (0.06) | 12 (0.06) | 3 (0.03) | 6 (0.03) | 1 (0.01) | 2 (0.01) |

Concordance: Both twins have the OA outcome in question, and both twins have the CVD outcome in question, i.e. they are similar for the outcomes in question both within- and cross-twin. *Card. arr*: cardiac arrythmia, *CHD* coronary heart disease, *MZ* monozygotic, *DZ* dizygotic. All MZ twins: $N = 9475$ pairs. All DZ twins: $N = 20\,510$ pairs.

**Table 3 | Correlations across any site osteoarthritis (OA), hip OA, knee OA and severe cardiovascular disease (CVD), respectively**

| OA | Adjusted for | Correlation between severe CVD and OA in same individual. Within-twin cross-trait, rPT (95% CI) | Correlation between severe CVD and OA across twins in an MZ pair. Cross-twin cross-trait, rMZ (95% CI) | Correlation between severe CVD and OA across twins in an DZ pair. Cross-twin cross-trait, rDZ (95% CI) |
|---|---|---|---|---|
| Any site OA | Crude | 0.16 (0.14–0.18) | 0.16 (0.13–0.19) | 0.11 (0.08–0.13) |
|  | Age and sex | 0.09 (0.08–0.11) | 0.08 (0.05–0.11) | 0.04 (0.03–0.05) |
|  | Age, sex and BMI | 0.07 (0.04–0.10) | 0.07 (0.05–0.09) | 0.03 (0.02–0.05) |
| Hip OA | Crude | 0.17 (0.15–0.19) | 0.20 (0.16–0.23) | 0.12 (0.09–0.14) |
|  | Age and sex | 0.07 (0.04–0.09) | 0.06 (0.03–0.10) | 0.03 (0.01–0.05) |
|  | Age, sex and BMI | 0.05 (0.03–0.08) | 0.06 (0.02–0.09) | 0.03 (0.01–0.05) |
| Knee OA | Crude | 0.15 (0.13–0.17) | 0.14 (0.11–0.17) | 0.09 (0.07–0.11) |
|  | Age and sex | 0.10 (0.08–0.12) | 0.07 (0.04–0.11) | 0.04 (0.02–0.05) |
|  | Age, sex and BMI | 0.08 (0.05–0.10) | 0.06 (0.03–0.09) | 0.03 (0.01–0.05) |

*rPT* phenotypic correlation, *rMZ* correlation in monozygotic twins, *rDZ* correlation in dizygotic twins, *Severe CVD* cardiac arrythmia, coronary heart disease, heart failure and/or stroke, *BMI* Body Mass Index.

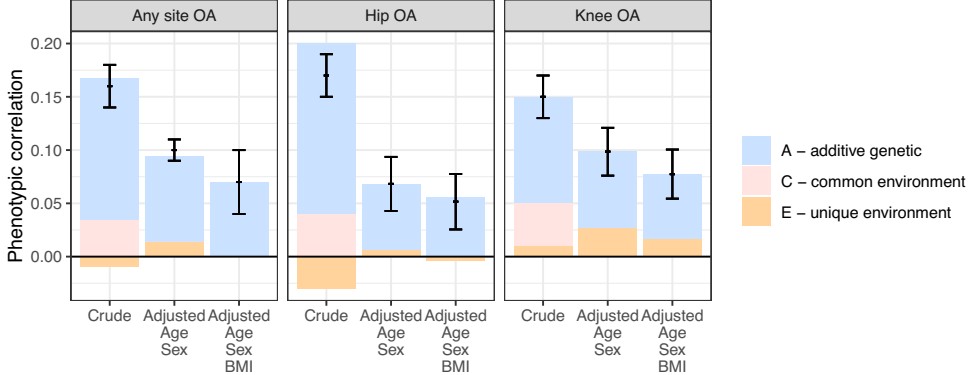

**Fig. 1 | Phenotypic correlations between site-specific OA and any type of severe cardiovascular disease.** Phenotypic correlations (black bars, i.e. correlations between hip osteoarthritis (OA) and knee OA, respectively, and severe cardiovascular disease in one individual) with 95% confidence interval, and parts of this correlation that are explained by additive genetic factors, common- and unique environmental factors. Analyses are performed separately by joint site, on 29 985 twin pairs, where 145 had any site OA (hip, knee and/or hand OA) and any of CVDs, 38 had hip OA and any of CVDs and 44 had knee OA and any of CVDs (pair level). Please note that the total correlation (denoted with black bars) can be lower than the sum of A and C components, when the E component has negative value. Data are presented as correlations +/− 95% confidence intervals. Source data are provided as a Source Data file.

strong contribution to a shared etiology from genetic factors (S-Table 8).

## Hip OA vs Knee OA and any of severe CVD

Pairs concordant for both hip OA and any of CVDs were generally older, consisted of more females and had lower BMI than pairs concordant for both knee OA and any of CVDs (Table 1). We observed minor differences in the crude and age- and sex-adjusted phenotypic correlation between any of CVDs and OA at the hip vs knee joint (Table 3, Fig. 1). For hip OA, additive genetic factors explained 86% (rPT due to A = 0.06 (95% CI = 0.03–0.10) of the variance, shared environmental factors 0% (rPT due to C = 0.00 (95% CI = −0.02–0.02), unique environmental factors 14% (rPT due to E = 0.01 [95% CI = −0.03–0.04]) in age- and sex-adjusted analyses (Fig. 1). For knee OA, additive genetic factors explained 70% (rPT due to A = 0.07 [95% CI = 0.04–0.10]) of the variance, shared environmental factors 0% (rPT

**Table 4 | Correlations across any site osteoarthritis (OA) and specific cardiovascular diseases (CVDs)**

| Specific CVD | Adjusted | Correlation between any site OA and CVDs in same individual Within-twin cross-trait, rPT (95% CI) | Correlation between any site OA and CVDs across twins in an MZ pair Cross-twin cross-trait, rMZ (95% CI) | Correlation between any site OA and CVDs across twins in an DZ pair Cross-twin cross-trait, rMZ (95% CI) |
|---|---|---|---|---|
| Cardiac arrythmia | Crude | 0.16 (0.14–0.18) | 0.18 (0.15–0.22) | 0.09 (0.08–0.11) |
|  | Age and sex | 0.12 (0.10–0.15) | 0.12 (0.09–0.16) | 0.06 (0.04–0.08) |
| CHD | Crude | 0.14 (0.12–0.16) | 0.16 (0.13–0.19) | 0.08 (0.06–0.10) |
|  | Age and sex | 0.10 (0.08–0.12) | 0.09 (0.06–0.12) | 0.05 (0.03–0.06) |
| Heart failure | Crude | 0.08 (0.05–0.11) | 0.05 (0.00–0.09) | 0.03 (0.01–0.06) |
|  | Age and sex | 0.05 (0.02–0.08) | 0.00 (-0.04–0.05) | 0.00 (-0.02–0.02) |
| Stroke | Crude | 0.09 (0.07–0.11) | 0.10 (0.07–0.14) | 0.06 (0.04–0.08) |
|  | Age and sex | 0.05 (0.02–0.07) | 0.04 (0.00–0.08) | 0.02 (0.00–0.04) |

*rPT* phenotypic correlation, *rMZ* correlation in monozygotic twins, *rDZ* correlation in dizygotic twins, *CHD* coronary heart disease.

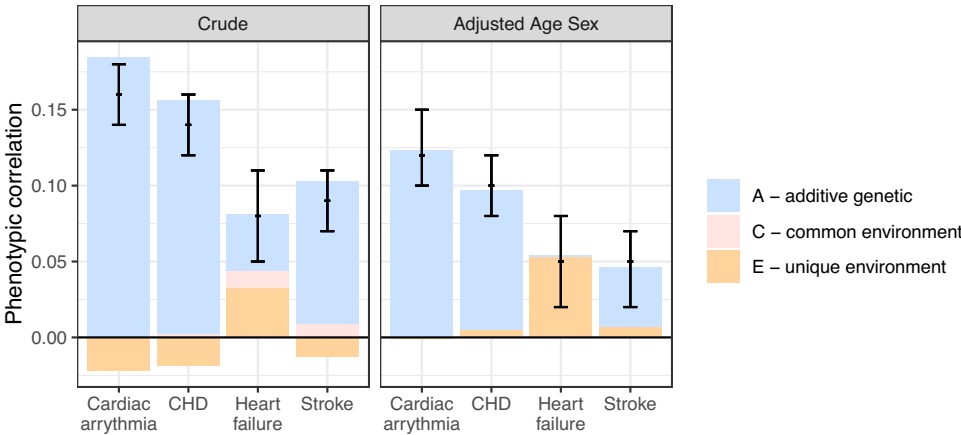

**Fig. 2 | Phenotypic correlations between any site OA and each of specific types of severe cardiovascular disease.** Phenotypic correlations (black bars, i.e. correlations between osteoarthritis and specific, severe cardiovascular disease (CVD) in one individual) with 95% confidence interval, and parts of this correlation that are explained by additive genetic factors, common- and unique environmental factors. Analyses are performed for any site OA (hip, knee and/or hand OA) combined with specific CVDs on 29 985 twin pairs, i.e. where 39 had cardiac arrythmias and any site OA, 49 had CHD and any site OA, 15 had heart failure and any site OA, and 18 had stroke and any site OA (pair level). Please note that the total correlation (denoted with black bars) can be lower than the sum of A and C components, when the E component has negative value. CHD: coronary heart disease. Data are presented as correlations +/− 95% confidence intervals. Source data are provided as a Source Data file.

due to C = 0.00 [95% CI = −0.02–0.02]), unique environmental factors 30% (rPT due to E = 0.03 [95% CI = −0.00–0.06]) in age- and sex-adjusted analyses (Fig. 1). The respective proportions explained by genetic factors vs environmental factors were largely similar with and without adjustment for BMI (Fig. 1, i.e. -100% of hip OA/CVD association explained by shared genetic liability, compared to -75% of knee OA/CVD association explained by shared genetic liability). We did not explore sex differences due to low numbers (S-Table 3).

**Any site OA and specific CVDs**
For cardiac arrythmias and CHD (including acute myocardial infarction, valvular disease, and angina pectoris), we found a similar pattern as for any of the severe CVDs. The phenotypic correlations were low and the cross-twin-cross-trait correlation equally high as the phenotypic correlation, yet slightly higher than the DZ cross-twin-cross-trait correlation (Table 4). Again, this pattern suggests genetic factors are of importance in the shared etiology of cardiac arrythmias and any-site OA as well as CHD and any-site OA, respectively. Results suggest that up to 90–100% of the correlation between OA and cardiac arrythmias, and between OA and CHD can be explained by genetic factors when adjusted for age and sex (Fig. 2).

The phenotypic correlations between any-site OA and heart failure, and any-site OA and stroke were lower than observed for cardiac

arrhythmia and CHD (Table 3, Fig. 1). There were also few differences between the rPT, rMZ and rDZ correlations for these traits (Table 3), which might be explained by the low numbers of concordant pairs (Table 2). In ACE analyses, the phenotypic correlation between any-site OA and heart failure could to a low extent, be explained by genetic factors (A part): The contribution of additive genetic and shared environmental factors was close to zero (rPT=0.00 [95% CI = −0.05–0.05] [0%] and rPT=0.00 [95% CI = −0.02–0.10] [0%], respectively), i.e. unique environmental factors explained rPT=0.05 (95% CI = 0.00–0.10) 100% of the correlation in age- and sex-adjusted analyses. We did not explore sex differences due to low numbers (S-Table 3).

**Impact of genetic vs observed shared environmental risk factors on bivariate concordance rate for any site OA and any of the severe CVDs**
In general, being similar on risk factors was more common in MZ twins (*N* = 9475) than in DZ twins (*N* = 20,510) (Table 5). The MZ pairs that were concordant for OA and CVD were more often sharing their low education than all MZ pairs (75% vs 61%) and also their overweight (39% vs. 16%) (Table 5). A similar pattern could be seen in DZ pairs (70% vs 61% and 38% vs 15%, respectively) (Table 5). We observed fewer differences for risk factors smoking and physical inactivity (Table 5).

**Table 5 | Descriptive characteristics showing degree of shared observed environmental factors, pair level**

| | All MZ pairs | MZ Pairs where both twins have both OA and severe CVD | All DZ pairs | DZ Pairs where both twins have both OA and severe CVD |
|---|---|---|---|---|
| | **N = 9475** | **N = 64** | **N = 20 510** | **N = 81** |
| Age at start of follow-up, mean (SD) | 46 (17) | 60 (9) | 49 (16) | 61 (10) |
| Same sex, M, n (%) | 4128 (44) | 34 (53) | 5400 (26) | 27 (33) |
| Same sex, F, n (%) | 5347 (56) | 30 (47) | 6617 (32) | 26 (32) |
| Opposite sex, n (%) | 0 | 0 | 8493 (41) | 28 (35) |
| College/university (>1 years), n (%) | | | | |
| Both twins high education | 2258 (24) | 10 (16) | 3069 (15) | 6 (7) |
| One twin high, one twin low education | 1474 (16) | 6 (9) | 4835 (24) | 18 (22) |
| Both twins low education | 5743 (61) | 48 (75) | 12,606 (61) | 57 (70) |
| Overweight, ≥25 kg/m2, n (%) | | | | |
| None of twins overweight | 5799 (62) | 26 (41) | 9336 (46) | 24 (30) |
| One twin overweight, one twin normal weight | 1543 (16) | 13 (20) | 6376 (31) | 22 (27) |
| Both twins overweight | 1478 (16) | 25 (39) | 3040 (15) | 31 (38) |
| Ever regular smoker, n (%) | | | | |
| None of twins smokers | 2931 (31) | 21 (33) | 5253 (26) | 20 (25) |
| One twin smoker, one twin non-smoker | 1906 (20) | 17 (27) | 6612 (32) | 37 (46) |
| Both twins smokers | 4291 (45) | 26 (41) | 7795 (38) | 24 (30) |
| Physical inactivity, n (%) | | | | |
| Both twins physically active | 2017 (22) | 13 (22) | 2214 (16) | 9 (17) |
| One twin physically ative, one twin physically inactive | 2051 (23) | 17 (28) | 4005 (30) | 14 (26) |
| Both twins physically inactive | 3737 (42) | 21 (35) | 5103 (38) | 19 (35) |

*OA* osteoarthritis, *CVD* cardiovascular disease, *CHD* coronary heart disease, *SD* standard deviation, *MZ* monozygotic, *DZ* dizygotic. *BMI data is missing for 8.05% of study population, smoking data is missing for 3.9% of study population, physical activity data is missing for 36% of the study population.

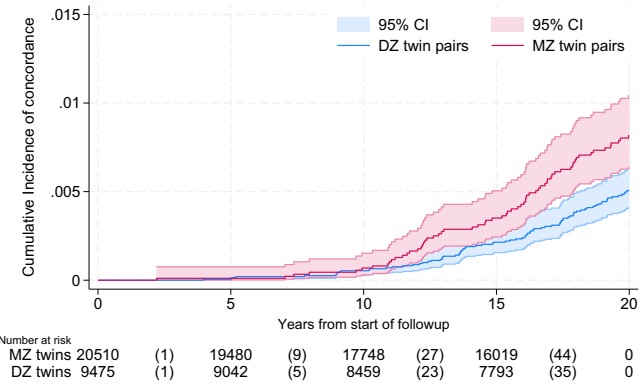

**Fig. 3 | The cumulative incidence of bivariate concordance for OA and CVD.** The cumulative incidence of bivariate concordance for OA and CVD from the start of follow-up (time 0, Jan 1st 1997) until the second twins get the last diagnosis of OA or CVD (event). MZ: monozygotic, DZ: dizygotic. Data are presented as the cumulative incidence +/− 95% confidence intervals. Source data are provided as a Source Data file.

The cumulative incidence of becoming concordant for both OA and severe CVD was higher for MZ than for DZ pairs (Fig. 3). In Cox regression analyses studying the number of years from start of follow-up, being an MZ pair was associated with an increased risk of concordance for OA and severe CVD when compared to DZ pairs also after controlling for lifestyle factors (Table 6), confirming the relevance of shared genetic factors impacting on the disease. However, shared overweight, i.e. when both twins had BMI above 25 kg/m2 gave a similarly increased risk, implying that overweight may be a shared environmental risk factor influencing co-occurrence of OA and severe CVD to a similar extent as genetics.

Interestingly same-sex twins had a higher risk than opposite-sex twins, and as expected, higher age gave higher risk of concordance (Table 6). Shared smoking and physical inactivity did not result in increased risk (Table 6). However, data on physical activity level should be interpreted with care due to low numbers, i.e. we did not include this risk factor in the multivariate analysis as it would lead to lower precision for all estimates. As a replacement, physical activity level was included together with zygosity in a reduced multivariate model, i.e. where they were adjusted for each other, showing HR = 1.16 (95% CI = 0.67–2.01) for pairs where one twin is physically active and the other is inactive, and HR = 1.08 (95%CI = 0.64–1.82) for pairs where both are physically inactive. In comparison in this analysis, the risk for MZ twins compared to DZ twins remained high, at HR = 1.71 (95% CI = 1.14–2.58).

## Discussion

In this study of almost 60 000 twins, we report that a major part of the correlation between OA and severe CVD in an individual can be explained by genetic factors that are shared for the two types of diseases. A shared genetic liability could be of greater importance for the correlation between hip OA and any of severe CVDs than for knee OA. For knee OA and any of severe CVDs, we found some evidence supporting importance of unique environmental factors. However, the adjustment for age, sex, education level, BMI/overweight, smoking and physical activity level did not attenuate the strong genetic liability for shared OA and severe CVD. Among the included specific CVDs, the shared genetics were seen for OA and cardiac arrythmias, OA and CHD, and OA and stroke, whereas unique environmental factors may play a larger role for co-occurrence of OA and heart failure. However, for the latter, findings are more uncertain due to few cases. Our results challenge the persisting opinion that OA is a risk factor for severe CVD, with heart failure being a potential exception.

**Table 6 | Cox regression analyses showing relevance of proxies for genetics vs environmental risk factors**

| | Univariate Cox regression analyses N = sample sizes a, b, c, d in respective analysis<br>HR (95%CI) | Multivariate Cox regression analyses N = 27 458 of which 141 with outcome<br>HR (95%CI) |
|---|---|---|
| Zygosity a | | |
| DZ twins | 1 (ref) | NA |
| MZ twins | 1.62 (1.17–2.25) | NA |
| Sex a | | |
| Opposite sex DZ twins | 1 (ref) | 1 (ref) |
| Same sex MZ twins, male | 2.50 (1.52–4.13) | 3.64 (2.14–6.19) |
| Same sex MZ twins, female | 1.69 (1.01–2.82) | 2.34 (1.34–4.07) |
| Same sex DZ twins, male | 1.64 (0.97–2.79) | 2.05 (1.18–3.55) |
| Same sex DZ twins, female | 1.30 (0.76–2.22) | 1.40 (0.79–2.49) |
| Age (always shared) a | 1.11 (1.10–1.13) | 1.11 (1.10–1.13) |
| College/university (>1 years) a | | |
| Both twins high education | 1 (ref) | 1 (ref) |
| One twin high, one twin low education | 1.34 (0.71–2.52) | 0.95 (0.50–1.79) |
| Both twins low education | 2.44 (1.44–4.12) | 0.89 (0.52–1.52) |
| Overweight, ≥25 kg/m2 b | | |
| None of twins overweight | 1 (ref) | 1 (ref) |
| One twin overweight, one twin normal weight | 1.43 (0.93–2.21) | 1.40 (0.90–2.19) |
| Both twins overweight | 4.21 (2.88–6.17) | 3.44 (2.31–5.12) |
| Ever regular smoker c | | |
| None of twins smokers | 1 (ref) | 1 (ref) |
| One twin smoker, one twin non-smoker | 1.20 (0.80–1.80) | 1.77 (1.16–2.71) |
| Both twins smokers | 0.75 (0.50–1.14) | 1.37 (0.88–2.12) |
| Physically active d | | |
| Both twins physically active | 1 (ref) | Omitted due to missing data |
| One twin physically ative, one twin physically inactive | 1.08 (0.62–1.86) | |
| Both twins physically inactive | 1.06 (0.63–1.78) | |

*HR* Hazard Ratio, *CI* confidence intervals, *OA* osteoarthritis, *CVD* cardiovascular disease, *CHD* coronary heart disease, *MZ* monozygotic. *DZ*: dizygotic. A: N = 29,985 of which 145 with outcome. B: N = 27,572 of which 141 with outcome. C: N = 28,788 of which 145 with outcome. D: N = 19,127 of which 93 with outcome.

## Comparison to previous studies

To our knowledge, no previous studies have examined any potential shared pathways between OA and severe CVD using a bivariate twin design. Our findings shed important new light to the existing literature considering the association between OA and severe CVD[16]. For example, our findings may fully or partly explain the 24% to 69% increased risk of CVD in OA patients compared to the general population or matched non-OA cohorts as found in two meta-analysis of observational studies[1,15]. The increased risk of CVD in OA patients has previously been explained by shared risk factors such as obesity, high age, hypercholesterolemia, and diabetes[17–19], but also by consequences of joint pain, such as physical inactivity and NSAID use, which both are independent risk factors for CVD[20]. In the bivariate twin analyses, we adjusted for age, sex and the average BMI in adult life. In Cox regression analyses, we additionally adjusted for education level, smoking and physical inactivity. Together, the findings help to understand the extent to which the observed associations between genetic/environmental factors and the phenotype are influenced by these variables. With adjustment, the phenotypic correlations were typically lowered, yet the high proportion (around 70% to 100%) of OA/CVD correlation explained by genetic factors was similar as in crude analyses. The negative correlation explained by unique environmental factors in crude analyses (Fig. 1, Fig. 2) may be due to an opposite effect on the two outcomes of interest, for example, high level of sports participation could increase risk of injury and thus OA but decrease risk of CVD. Similarly, neither overweight, age, sex, education, smoking or physical activity level attenuated the Hazard Ratios in Cox regression analyses (Table 6), implying the strong genetic effect may be independent of influence from typical environmental lifestyle factors that typically increases the risk of both OA and CVD. A strong genetic effect for combined OA and CVD was seen for both sexes in both analyses, however the two diseases seem to co-occur more often in males than in females (S-Table 6).

## Clinical relevance and interpretation

The new insight suggests that OA may not be a causal risk factor for CVD. Instead, common genetic liabilities may imply shared biological pathways between OA and CVD. In similar ways as previously suggested in reviews of similar studies of psychiatric diseases[21], such genetic overlap between two disorders may suggest that the current diagnostic boundaries between OA and severe CVD may need to be reevaluated. If our findings are confirmed in other samples and with other research methodology, there may be evidence for a distinct OA-CVD phenotype. A potential OA-CVD phenotype has previously been hypothesized[16,22] and would need to be thoroughly described from a research perspective both within the field of OA and within the field of CVD. For example, future efforts should investigate whether there are any specific genes or genetic markers that might explain the shared heritability. These efforts should also further explore the role of overweight or obesity in modifying genetic influence. Possibly, overweight/obesity, but also other lifestyle factors are inherited directly through genes from parents or indirectly through parental influence

on the environment. For example, we and other research groups have previously reported that parental OA is a risk factor for offspring's OA[23], as well as parental CVD is a risk factor for offspring's CVD[24], underlining the need for physicians to examine the family history of OA and/or CVD. Beyond the new insights in disease etiology, the knowledge gained from this study may aid in the earlier detection or prevention of comorbid disease between OA and CVD. If findings are confirmed and more detail to these associations are provided in future research, tailored or personalized medicine approaches can be developed with the aim of preventing severe disease and disability.

## Strengths and limitations

Important strengths of our study are the inclusion of a large, population-based twin sample with limited selection bias and high degree of generalizability to other countries with similar healthcare system as Sweden. Another strength is the inclusion of several OA and CVD outcomes and the application of a similar twin modelling strategy across the outcomes. This approach ensures simplicity and comparability and can serve as a basis for estimating more complex genetic and environmental mechanisms.

However, it should be noted that we cannot conclude regarding the temporality of events in the current study. We included prevalent diagnoses and did not consider whether OA occurred prior to CVD or vice versa. There may be important information stored in the timing of events, i.e. strong genetic correlations could be reflected in shorter time passing in between the events, which to some extent, is reflected in Fig. 3. Further, CVD may be a cause of death, preventing individuals from developing OA if the CVD-related death occurred at early age. Our sensitivity analyses, including pairs where both twins survived through their entire possible follow-up period showed elevated crude estimates but similar age- and sex-adjusted estimates (S-Table 8). Given the younger age of individuals due to the conditioning on survival in our sensitivity analyses, these findings are as expected, and future studies of genetic influence should further explore the interaction between age and timing of OA and CVD events.

Another limitation might be that the self-reported height, weight, smoking and physical activity data in our study were measured at different ages and over a long time period (1963-2012). We had missing BMI-data for 8% of the total sample and missing physical activity data for 30% of the sample. We did not consider multiple imputation because the imputation model would need to adequately capture the additional complexity that is present in twin models, and a misspecification can lead to biased estimates. Also, BMI was missing in low proportion of twin pairs, however data on physical activity for a high proportion, and analyses including physical activity level should be interpreted with care. Our study may be further limited due to the use of data only from specialist care, while many OA patients are typically seen in primary care. However, given that we do not consider the timing of diagnoses, it is enough that a person consults specialist care only once to be captured. The sensitivity of specialist care diagnosis in years 2000 to 2016 to capture patients with OA diagnosed at any care level in southern region of Sweden (Skåne) is 66% for knee OA, 72% for hip OA and 62% for hand OA, suggesting that the majority of patients consulting for their OA would be included in this study (unpublished data). Another limitation may be the low numbers with hand OA, preventing the study of hand OA and severe CVD as an own phenotype. And finally, we cannot rule out violations to the equal environment assumption (EEA), which is a common critique against twin studies based on the classical twin design when twins are reared together[4]. However, the descriptive data of degree of shared environments imply that MZ twins share their low education, overweight etc to an equal extent as DZ twins (Table 5), supporting there may not be large observable violations to the EEA.

In summary, the correlation between OA and severe CVD in the same individual was rather low, however, a major part of it could be

explained by shared genetic factors, for both males and females. Our findings suggest that genetic factors that are shared for OA and CVD may contribute to both diseases, independent of other strong contributors like age, body weight, education level and lifestyle factors. These interpretations challenge the idea that OA is predominantly a risk factor for CVD or vice versa.

## Methods

### Inclusion and ethics

This research complies with all relevant ethical regulations. Inclusion procedures for the Swedish Twin Registry are described at https://ki.se/en/research/swedish-twin-registry-for-researchers. The study was approved by the Ethical Review Board at Lund University, Sweden. Participants gave written informed consent to be included in the Swedish Twin Register, which includes consent to linkage to other registries. Participants were not compensated for their participation.

### Data and design

We used data from the Swedish Twin Registry (STR) - a national twin registry, including more than 194,000 Swedish twins, i.e. covering 95–100% of all twins born in Sweden[25]. We linked the STR to the National Patient Register (NPR) in 2017 and extracted diagnostic codes of musculoskeletal disease and cardiovascular disease obtained in specialist care from 1997 and onwards, including data on every hospitalization (from 1997 to 2016), 1-day surgery (from 1997 to 2016) and specialist outpatient care visit (from 2001 to 2016). We also linked with the National Board of Health and Welfare, including the longitudinal integration database for health insurance and labour market studies (LISA) from Statistics Sweden for retrieval of demographic variables, and the Cause of Death register for retrieval of deaths. Zygosity was determined based on DNA testing (13%) or questions about intra-pair physical similarities in childhood[25]. We included twins aged ≥18 years in 1997 with a minimum follow-up time of 1 year in the NPR. Follow-up began in 1997. We required both twins in a pair to be alive for at least the first year of follow-up.

### Outcomes

We studied the diagnoses OA in knee, hip or hands and selected CVDs as reported in the NPR according to the Swedish translations of ICD-10 from 1997 to 2017 (S-Table 1). We defined any site OA as the first diagnosis of OA in specialist care in the hip, knee and/or hand joint. Thus, as an example, if one twin in a pair had a diagnostic code M17 (OA of the knee joint) and the other twin had a diagnostic code M16 (OA of the hip joint), both twins would be categorized as having the outcome any site OA. Similarly, we defined severe CVDs as the first diagnosis of the following CVDs: cardiac arrhythmias, coronary heart disease (CHD) including acute myocardial infarction, valvular disease and angina pectoris), heart failure and stroke (hemorrhagic and ischemic) in specialist care, again according to the Swedish translations of ICD-10 (S-Table 1). Thus, if one twin in a pair had diagnostic code I21 (acute ischemic heart attack) and the other twin had a diagnostic code I50 (heart failure), both twins would be categorized as having the outcome of severe CVD. We also studied site-specific OA (hip joint vs knee joint) and each of the specific severe CVD categories (S-Table 1). Using these outcomes, the twins were required to have the codes from the specific OA or CVD category. In case of multiple diagnostic codes within the same category for an individual (S-Table 1), we selected the first recorded diagnostic code after start of follow-up. We allowed for having multiple of the diagnostic categories as described in S-Table 1. All hospitals had access to the same national guidelines for diagnostics of OA and severe CVD in accordance with the ICD-system, provided by the National Board of Health and Welfare[26]. However, there may be differences in diagnostic methods across hospitals. These differences are unlikely to bias the results in this study as we assume that they affect identical vs non-identical twin pairs in a random manner.

### Observed covariates potentially important for both OA and severe CVD

Age, sex, height in centimeters and weight in kilograms as well data on smoking and physical activity level were self-reported on questionnaires in years 1963, 1967, 1970, 1973, 2000, 2006 and/or 2012[25]. Body mass index (BMI) was calculated for each of these time points as $kg/m^2$ and averaged over all available measurements. Data on smoking were reported as being a current or former regular smoker and participants were categorized as being an ever regular smoker (value 1) or not (value 0, ref. category). Data on physical activity level was measured on different scales ranging from binary scales to scales from 0 to 5, 7 or 10, where a higher number indicated a higher physical activity level, longer lasting and/or with higher intensity (S-Table 2). Data were synthesized to form a binary variable (based on 1/3rd highest values being coded as physically active (value 0) and 2/3rd lowest as inactive (value 1). This variable is likely to indicate regular physical activity during leisure time, likely with moderate intensity leading to sweating and/or higher perspiration rate for at least 30 min, yes or no. Data on education level (primary- and secondary school vs college/university level) were collected from LISA, Statistics Sweden.

### Statistical analyses

First, we described individuals/pairs having combinations of OA and severe CVD on their background characteristics/observed covariates and also the numbers of concordant pairs. Concordant twin pairs in our study are pairs in which both twins have received both the diagnoses of interest. If genetic effects are important, we would expect monozygotic (MZ) twins to be more concordant for both diagnoses than dizygotic (DZ) twins.

After descriptive analyses, we applied bivariate twin models including OA (any-site, and hip OA vs knee OA separately), any of severe CVDs and each of the four severe CVDs (S-Table 1), in separate analyses. With these models, one can estimate whether shared or genetic causes vs unique environmental causes can explain the prevalence of OA and the different CVDs in the same individual, when averaged over all individuals[27,28]. We calculated two types of correlations: within-twin between-trait correlations, often referred to as "phenotypic correlation" (i.e. rPT, the occurrence of two diseases within an individual) with 95% confidence intervals (CIs), and cross-twin cross-trait correlations (i.e. the occurrence of two diseases in two twins) with 95% CI. The latter involved one twin in a pair having osteoarthritis (OA) while the other twin had one of the CVDs. A greater cross-twin cross-trait correlation in MZ twins compared to DZ twins suggests that shared genetic factors contribute to the observed relationship between the two traits. Conversely, if the MZ cross-twin cross-trait correlation is lower than the phenotypic correlation, it indicates the presence of unique environmental factors influencing each of the diseases separately. To further dissect the phenotypic correlation, we examined the portion of it attributable to genetic (A), common environmental (C), and unique environmental (E) factors, again with 95% CIs. The latter analyses are based on the knowledge that MZ twins share 100% of their genes, while DZ twins share on average 50% of their genes. Within the bivariate twin model (S-Fig. 1), the phenotypic correlation (rPT) due to genetic effects (A) was calculated based on individual heritabilities for each trait ($\sqrt{A1}$ and $\sqrt{A2}$) and the genetic correlation between these traits (Ra), (i.e. [$\sqrt{A1} * Ra * \sqrt{A2}$)/rPT]). This proportion may also be called bivariate heritability[29]. Due to our many outcomes and to optimize possibilities for comparison across combination of traits, we applied ACE-models only (i.e. we did not consider other model specifications like the ADE-model, CE-model etc). Thus, an underlying assumption of our analyses is that genetic and environmental influences are additive and do not interact in complex ways. Further, because measures of univariate heritability for OA vs severe CVD separately are previously published based on the Swedish Twin Registry[6,10,11,13,14], we will focus on presenting measures in the bivariate

scenario, i.e. separating the total phenotypic correlation between OA and severe CVD into parts due to A, C and E as explained above.

All analyses were run crude and adjusted for age and sex. In addition, we assessed to what extent BMI—as a shared risk factor for OA (any-site, hip OA and knee OA, respectively) and any of the severe CVDs—impacted on the results. We explored potential sex differences in the correlation between OA and severe CVDs by stratifying the analyses on male twin pairs and female twin pairs as well as analyses of opposite-sex DZ twins only vs same-sex DZ twins only.

To our knowledge, bivariate twin models for binary outcome data cannot take censoring into account, meaning that any individual or pair that typically would have been censored due to death in a prospective cohort study would need to be classified as true diagnosed or true non-diagnosed cases in the present study. Thus, all analyses are of prevalent rather than incident diagnoses, meaning that if an individual died with the outcome, he/she would be classified as diagnosed and if an individual died without the outcome, he/she would be classified as non-diagnosed in our analyses. As such, bivariate twin models for binary outcome data may include pairs that, prospectively seen, do not have the chance to become a concordant pair for one or two of the traits (for example, due to death in an undiagnosed twin prior to a co-twin being diagnosed). Importantly, we stopped the follow-up time for both twins in a pair, at the time of the earliest death in that pair. Also, the censoring in our study is mainly administrative, as the follow-up is done through a national healthcare register.

To estimate whether results are sensitive to this impossibility to model the data prospectively, we repeated the main analyses (any site OA and any of CVDs) in the selection of pairs who survived through the entire follow-up (i.e. conditioning on survival). For all analyses of unobserved variance factors (genetics vs environment), we used a bivariate constrained polychoric correlation model in which thresholds were equated across MZ and DZ group and phenotypic correlations were equated across twin order[28]. Further, to investigate whether genetic factors impacted on the rate of becoming concordant for OA and severe CVD over time, and also, to examine whether shared environmental factors could modify this relationship, we performed exploratory/descriptive Cox regression analyses. The analysis unit in this model was a twin pair (not an individual twin). The start of follow-up time was as previously (as both twins enter the study at the same time point), while the event of interest was concordance of the two twins in a pair on presence of both OA and CVD (i.e. the date at which both twins in a part developed both OA and CVD). Pairs who did not become concordant were censored at study end or death of at least one twin in a pair, whichever came first. In these analyses, we first included zygosity (DZ = 0 and MZ = 1) as the independent variable (i.e., a proxy for genetic factors), univariately. In a multivariate analysis, we also included sex (coded by zygosity, i.e. same-sex MZ males, same-sex MZ females, same-sex DZ males, same-sex DZ females, vs opposite-sex DZ pairs (ref. category) and twin pair age at start of follow-up. Finally, education level, overweight (based on BMI ≥ 25 kg/m2 yes (1) or no (0, ref. category), smoking and physical activity level were included as proxies for shared environmental factors in this multivariate analysis. To do this, we calculated the mean value of the two twins. As an example, for the binary variable smoking, we would construct a three-level variable indicating 1) concordant for non-smoking, 2) discordant for smoking, and 3) concordant for smoking. The multivariate analyses were performed without having any predefined assumptions of causal structure. RStudio v2023.06.2, OpenMX: Extended Structural Equation Modelling v2.21.11 (27) were used for all bivariate twin model analyses. StataNow/MP 18.5 for Windows (64-bit x86-64) were used for describing data and for Cox regression analysis.

### Reporting summary

Further information on research design is available in the Nature Portfolio Reporting Summary linked to this article.

## Data availability

The dataset of this study was the Swedish Twin Registry, which is a property of Karolinska Institutet, provided to the researchers through a restricted-access agreement that prevents sharing the dataset with a third party or publicly. Individual-level data of patients included in this paper after de-identification are considered sensitive and will not be shared. However, the individual-level data in the Swedish Twin Register are accessible to authorized researchers after ethical approval and application to "https://ki.se/en/research/swedish-twin-registry-for-researchers" administered by Karolinska Institutet. Data requests may be sent using the web application available at the STR website. Source data are provided with this paper.

## Code availability

Computer codes can be found in the attached supplementary file (S-Methods 1 and S-Methods 2).

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

## Acknowledgements

We would like to acknowledge The Swedish Twin Registry and Barbro Sandin and Patrik Magnusson for access to the data and data management. The Swedish Twin Registry is managed by Karolinska Institute and receives funding through the Swedish Research Council under the grant no 2017-00641. This study was funded by the Swedish Research Council (E0234801), the Greta and Johan Kock Foundation, the Swedish Rheumatism Association, the Österlund Foundation, Governmental Funding of Clinical Research within the National Health Service (ALF), and the Foundation for People with Movement Disability in Skåne. The funding sources had no influence on the design or conduct of the study, the collection, management, analysis, or interpretation of the data, the preparation, review, or approval of the manuscript, or the decision to submit the manuscript for publication.

## Author contributions

Karin Magnusson had full access to all of the data in the study and take responsibility for the integrity of the data and the accuracy of the data analysis. *Concept and design:* Magnusson, Turkiewicz *Acquisition, analysis, or interpretation of data:* All authors. *Drafting of the manuscript:* Magnusson, Dell'Isola *Critical revision of the manuscript for important intellectual content:* All authors *Statistical analysis:* Magnusson, Turkiewicz *Obtained funding:* Englund *Administrative, technical, or material support:* N/A *Supervision:* N/A All authors contributed in drafting the article or critically revising it for important intellectual content. All authors gave final approval for the version to be submitted.

## Funding

## Competing interests

We declare no conflicts of interest, except for Dr. Englund reporting consultancy for Grünenthal Sweden AB, and Key2Compliance AB (unrelated to current work).
