## [Peer Review File · Nature Communications]

REVIEWER COMMENTS

Reviewer #1 (Remarks to the Author):

This study is a unique investigation that utilizes the Swedish Twin Registry (STR), a national twin registry comprising over 194,000 Swedish twins, covering 95-100% of all twins born in Sweden, to evaluate the association between osteoarthritis and severe cardiovascular disease. The utilization of twin information and the strength of accessing a national database are commendable aspects of this research. However, this study has several critical weaknesses in its specific methodology.

Methods

1. Outcomes

Are the diagnostic methods for severe CVDs standardized across the hospitals participating in this study? Particularly, I think there is a tendency for variability in the diagnosis of heart failure. I speculate that there is variability between hospitals regarding whether heart failure with preserved ejection fraction (HFpEF) due to left ventricular diastolic dysfunction is included in the diagnosis of heart failure. How do you ensure consistency between hospitals? It would be necessary to clarify diagnostic criteria not only for heart failure but also for cardiac arrhythmias, coronary heart disease (CHD) including acute myocardial infarction, valvular disease, angina pectoris, and stroke.

2. Observed covariates potentially important for both OA and severe CVD

Environmental factors and lifestyle-related disease that closely influence severe CVDs cannot be ignored. In addition to smoking, exercise, DM, dyslipidemia, HT and diet, differences in the status of oral anti-arteriosclerosis drugs can significantly change the outcome.

Reviewer #2 (Remarks to the Author):

Remarks to the Author

In the manuscript entitled "Osteoarthritis and severe cardiovascular disease - a shared etiology?", based on the data from the Swedish Twin Registry, the authors applied bivariate twin models to examine the potential shared pathways between OA and severe CVD. This is an interesting and meaningful study that brings novel insights into the disease etiology. The manuscript is generally well-written and clear. I have the following comments for the authors to consider:

1. Please also provide all twin individual-level descriptive characteristics by OA joint site and sex-specific numbers of concordance for each outcome.
2. The authors mentioned that the most important known environmental risk factors for CVD include obesity, tobacco use and physical inactivity, which are also risk factors for OA. Therefore, it is also of interest to determine the extent to which physical activity and smoking, as shared risk factors for OA and any serious CVDs, impact the results.
3. It would be highly beneficial if authors could provide heritability estimates, which will also aid readers in understanding the extent of genetic influence on these diseases. Please describe the methods used to compute heritability within the bivariate model, including any statistical assumptions implemented in

the analysis.

4. ACE models are well-reported. However, the readers can benefit from information about

a. Additional detail on the assumptions inherent to the twin model, specifically how equal environments and zygosity determination were handled.

b. Model fitness BIC, AIC and AUC.

5. The discussion effectively highlights key findings regarding shared genetic liability; however, it could be enhanced by integrating more on the biological mechanisms or pathways that might underlie these genetic correlations. Are there any specific genes or genetic markers identified that might explain the shared heritability? This would provide more depth and context to the genetic correlations observed.

6. If space allows, the authors might comment in the Discussion about the finding that unique environmental factors may play a larger role in the common cause of OA and HF. This might include, but not be limited to, lifestyle factors, medication use, or systemic inflammation that are particularly relevant to HF but not necessarily to other types of CVDs. Exploring these distinctions could provide valuable insights into not only the pathophysiology of these diseases but also potential prevention strategies.

Reviewer #2 (Remarks on code availability):

NA

Manuscript NCOMMS-24-22960A: "Osteoarthritis and severe cardiovascular disease - a shared etiology?"

We would like to thank the expert reviewers for their valuable input, which has helped to improve our manuscript. Please find below a point-to-point response to your comments and a list of the changes we made in the revised manuscript. The page and line references refer to the marked version of the manuscript.

Comments of reviewer 1	Our response	Action
This study is a unique investigation that utilizes the Swedish Twin Registry (STR), a national twin registry comprising over 194,000 Swedish twins, covering 95-100% of all twins born in Sweden, to evaluate the association between osteoarthritis and severe cardiovascular disease. The utilization of twin information and the strength of accessing a national database are commendable aspects of this research. However, this study has several critical weaknesses in its specific methodology.	Thank you for your summary and encouraging comments.	
Methods 1. Outcomes Are the diagnostic methods for severe CVDs standardized across the hospitals participating in this study? Particularly, I think there is a tendency for variability in the diagnosis of heart failure. I speculate that there is variability between hospitals regarding whether heart failure with preserved ejection fraction (HFpEF) due to left ventricular diastolic dysfunction is included in the diagnosis of heart failure. How do you ensure consistency between hospitals? It would be necessary to clarify diagnostic criteria not only for heart failure but also for cardiac arrhythmias, coronary heart disease (CHD) including acute myocardial infarction, valvular disease, angina pectoris, and stroke.	Thank you for this important question. We agree that more information on diagnostic methods for severe CVDs, and also for OA needs to be included. Healthcare in Sweden is organized by regions, and thus all hospitals within a region are expected to at least follow the same diagnostic criteria (an example from region Stockholm [in Swedish]: https://kunskapsstodforvardgivare.se/omraden/hjart-och-karlsjukdomar/behandlingsprogram/akut-hjartsjukvard/hjartsvikt/bakgrund-och-diagnostik-vid-hjartsvikt) Moreover, all hospitals have access to and are recommended to use the same guidelines for diagnostics and treatment, which are available at the websites of the National Board of Health and Welfare. Further, the national data on heart failure and its outcomes are followed through a National Quality Register (https://www.ucr.uu.se/rikssvikt/) to increase the equality across the country. One of the key quality indicators followed through the register is for example documentation of LVEF at index healthcare visit with both primary and hospital care performing over the target levels. Similar situation is for other severe CVDs, that are followed through national registries including guidelines (example in Swedish). Importantly, in our case, we are interested in any similarities and differences within twin pairs. Particularly, we study concordance for OA and CVD for identical vs non-identical pairs. As far as we can see, there are no reasons to believe that any hospital-based inconsistency in diagnostic methods would affect identical and non-identical twin pairs differently, and if there are differences, they would occur by chance.	We have added to the section “outcomes”, at p. 4: “All hospitals had access to the same national guidelines for diagnostics of OA and severe CVD in accordance with the ICD-system, provided by the National Board of Health and Welfare (14). However, there may be differences in diagnostic methods across hospitals. These differences are unlikely to bias the results in this study as we assume that they affect identical vs non-identical twin pairs in a random manner.”
2. Observed covariates potentially important for both OA and severe CVD Environmental factors and lifestyle-related disease that closely influence severe CVDs cannot be ignored. In addition to smoking, exercise, DM, dyslipidemia, HT and diet, differences in the status of oral anti-arteriosclerosis drugs can	We agree there might be several other observed covariates that are potentially important for both OA and severe CVD. However, we are concerned that including more observed covariates in the model will give large challenges with model convergence. Adding more covariates will require larger sample sizes to maintain adequate statistical precision, and the inclusion of more observed covariates in a model that primarily estimates unobserved covariance can make the model more	We have added an additional analysis where we explore the associations between specific observed environmental factors and the outcome (bivariate concordance for OA and CVD). In doing so, we assessed the time to concordance for OA and CVD within a twin pair, investigating zygosity, age, sex, overweight, education level, smoking and physical inactivity as potential risk factors. The environmental factors were implemented in the regression analysis both as proxies for environmental factors but also as

Manuscript NCOMMS-24-22960A: "Osteoarthritis and severe cardiovascular disease - a shared etiology?"

significantly change the outcome.	difficult to fit and interpret, potentially leading to unstable estimates. Thus, we think a different model, mainly consisting of the observed data as mentioned by the reviewer could serve to provide information regarding the relative importance of genetics vs environment. We have data on lifestyle variables that typically precede the development of chronic diseases, i.e. smoking and physical activity level were self-reported several times from 1960 and onwards and we believe these variables are of interest in an analysis looking into the impact of lifestyle itself, but also the impact of the extent to which the lifestyle is shared by twins in a pair. We are not able to further adjust for the presence of diabetes or hypertension due to two reasons: these diseases are mainly treated in primary care, and our data cover specialist and inpatient care; further, we do not have enough years of data preceding study start to allow adjustment for diseases that could be true causes of OA or severe CVD. We believe that investigating smoking and physical activity, which are known risk factors for the chronic conditions mentioned by the reviewer, should partially account for these conditions in the analysis.	proxies for the degree to which these factors were shared between twins in a pair. Details of the analyses are described at p. 4 and 6. Results are described at p. 14-17, in brief showing that shared overweight, but not shared smoking or physical activity may impact the genetic associations, but that the differences between MZ and DZ twin are clear even after the adjustments. And finally, we have discussed these findings at p. 18 including how these analyses shed light on potential violations to the equal environment assumption on p 20. After these revisions, we believe we are more confident in our summary of findings and conclusions, i.e. we have underlined that our finding of a shared genetic liability is likely independent of observed lifestyle factors.
Comments of reviewer 2	Our response	Action
In the manuscript entitled "Osteoarthritis and severe cardiovascular disease - a shared etiology?", based on the data from the Swedish Twin Registry, the authors applied bivariate twin models to examine the potential shared pathways between OA and severe CVD. This is an interesting and meaningful study that brings novel insights into the disease etiology. The manuscript is generally well-written and clear. I have the following comments for the authors to consider:	Thank you for your summary and encouraging comments.	
1. Please also provide all twin individual-level descriptive characteristics by OA joint site and sex-specific numbers of concordance for each outcome.	We agree.	Twin individual level descriptive characteristics are provided in Table 1. Further, we have added references to our previous work on heritability and site-specific OA, p. 6: “In total 7 363 individuals had OA in any joint (of which 938, 2 498 and 3 845 in the hand, hip and knee joints, respectively, and as previously described in site-specific analyses (6, 17, 18)).” We now make a more clear distinction between individual level and pair level characteristics, by the inclusion of Table 5. We have added a table showing sex-specific concordance for each outcome in the supplementary material, S-Table 3.
2. The authors mentioned that the most important known environmental risk factors for CVD include obesity, tobacco use and physical inactivity, which are also risk factors for OA. Therefore, it is also of interest to determine the extent to which physical activity and smoking, as shared risk factors	We agree. However, there are concerns regarding adding more observed covariates to the model, as described in the response to Reviewer 1’s comment #2. Thus, we have found an alternative way of including these covariates, again as explained in the response to Reviewer 1’s comment #2.	Please see action to Reviewer 1’s comment #2.

Manuscript NCOMMS-24-22960A: "Osteoarthritis and severe cardiovascular disease - a shared etiology?"

for OA and any serious CVDs, impact the results.		
3. It would be highly beneficial if authors could provide heritability estimates, which will also aid readers in understanding the extent of genetic influence on these diseases. Please describe the methods used to compute heritability within the bivariate model, including any statistical assumptions implemented in the analysis.	We agree that providing more information could aid readers in understanding. However, after the addition of more risk factors and an additional analysis as an action reviewer 1's comment #2 and reviewer 2's comments #2, our manuscript and work has increased in size, resulting in a necessity to limit the presentation of additional data.	We have added a description of how the heritabilities were computed, including statistical assumptions implemented in the analysis, to the methods section at p. 5: “Within the bivariate twin model (S-Figure 1), the phenotypic correlation (r_{PT}) due to genetic effects (A) was calculated based on individual heritabilities for each trait ($\sqrt{A1}$ and $\sqrt{A2}$) and the genetic correlation between these traits (R_a), (i.e. $[\sqrt{A1} * R_a * \sqrt{A2}]/r_{PT}$). This proportion may also be called bivariate heritability (17). Due to our many outcomes and to optimize possibilities for comparison across combination of traits, we applied ACE-models only (i.e. we did not consider other model specifications like the ADE-model, CE-model etc). Thus, an underlying assumption of our analyses is that genetic and environmental influences are additive and do not interact in complex ways. Further, because measures of univariate heritability for OA vs severe CVD separately are previously published based on the Swedish Twin Registry (6, 10, 11, 18, 19), we will focus on presenting measures in the bivariate scenario, i.e. separating the total phenotypic correlation between OA and severe CVD into parts due to A, C and E as explained above.” Because the univariate heritabilities are previously published, we refrain from presenting these for all our outcomes and all our analyses. However, we think an example, for one analysis and a visual presentation of the bivariate twin model in the supplementary file can aid readers in understanding the association between genetic influence to one vs both diseases. An example is now presented in the results section at p.10: “In age-, sex- and BMI-adjusted analyses, additive genetic factors explained up to 100% (95% CI=57%-143%) (Figure 1), as calculated from $[\sqrt{A1} * R_a * \sqrt{A2}]/r_{PT}$ (S-Figure 1): heritability for OA ($\sqrt{0.475}$)=0.689 * genetic correlation between OA and CVD=0.178 * heritability for CVD=($\sqrt{0.328}$)=0.572)/total phenotypic correlation=0.07. Measures of model fit are presented in S-Table 4.”
4. ACE models are well-reported. However, the readers can benefit from information about a. Additional detail on the assumptions inherent to the twin model, specifically how equal environments and zygosity determination were handled. b. Model fitness BIC, AIC and AUC.	We agree.	We have added to the methods section, p. 3: “Zygosity was determined based on DNA testing (13%) or questions about intra-pair physical similarities in childhood (13).” In the bivariate twin model, as pointed out in our discussion section, we cannot rule out that violations to the equal environment assumptions have occurred. However, we believe the additional data on risk factors for MZ and DZ pairs may shed light on the degree of shared environmental factors, please see Table 6 and our discussion of these analyses at p. 18 and 20. We have added measures of model fit to the online supplementary file, to be found in S-Table 3.
5. The discussion effectively highlights key findings regarding shared genetic liability; however, it could be enhanced by integrating more on the biological mechanisms or pathways that might underlie these genetic correlations. Are there any specific genes or genetic markers identified that might	We agree that more on the biological mechanisms could be discussed.	We have added to p. 18-19: “For example, future efforts should investigate whether there are any specific genes or genetic markers that might explain the shared heritability. These efforts should also further explore the role of overweight or obesity in modifying genetic influence. Possibly, overweight/obesity, but also other lifestyle factors are inherited directly through genes from parents or

Manuscript NCOMMS-24-22960A: "Osteoarthritis and severe cardiovascular disease - a shared etiology?"

explain the shared heritability? This would provide more depth and context to the genetic correlations observed.		indirectly through parental influence on environment. For example, we and other research groups have previously reported that parental OA is a risk factor for offspring's OA (28), as well as parental CVD is a risk factor for offspring's CVD (29), underlining the need for physicians to examine the family history of OA and/or CVD. Beyond the new insights in disease etiology, the knowledge gained from this study may aid in earlier detection or prevention of comorbid disease between OA and CVD. If findings are confirmed and more detail to these associations are provided in future research, tailored or personalized medicine approaches can be developed with the aim of preventing severe disease and disability."
6. If space allows, the authors might comment in the Discussion about the finding that unique environmental factors may play a larger role in the common cause of OA and HF. This might include, but not be limited to, lifestyle factors, medication use, or systemic inflammation that are particularly relevant to HF but not necessarily to other types of CVDs. Exploring these distinctions could provide valuable insights into not only the pathophysiology of these diseases but also potential prevention strategies.	We agree the association between OA and HF is interesting. However, unfortunately numbers are small and we think our manuscript has become rather massive after the revisions.	We politely refrain from taking any further action to this point.

REVIEWERS' COMMENTS

Reviewer #1 (Remarks to the Author):

The revisions have significantly improved the clarity, methodology, and overall presentation of the research. The revised paper provides valuable scientific insights.

Reviewer #2 (Remarks to the Author):

the authors almost addressed my concerns in the latest version. I have only one more comment for the authors to consider:

in the discussion section, the authors stated "Among the included specific CVDs, the shared genetics were seen for OA and cardiac arrhythmias, OA and CHD, and OA and stroke, whereas unique environmental factors played a larger role for co-occurrence of OA and heart failure."

I suggested the authors in the last round to explain more about the finding that unique environmental factors may play a larger role in the common cause of OA and HF. If the authors are not able to comment further on this finding as they mentioned in the response letter that there are very small number HF cases, I therefore suggest the authors tune down a little bit regarding the roles of unique environmental factors on co-occurrence of OA and heart failure.

Manuscript NCOMMS-24-22960A: "Osteoarthritis and severe cardiovascular disease - a shared etiology?"

We would like to thank the expert reviewers for their valuable input, which has helped to improve our manuscript. Please find below a point-to-point response to your comments and a list of the changes we made in the revised manuscript. The page and line references refer to the marked version of the manuscript.

Comments of reviewer 1	Our response	Action
The revisions have significantly improved the clarity, methodology, and overall presentation of the research. The revised paper provides valuable scientific insights.	Thank you for your thorough review and encouraging comments.	
Comments of reviewer 2	Our response	Action
the authors almost addressed my concerns in the latest version. I have only one more comment for the authors to consider: in the discussion section, the authors stated "Among the included specific CVDs, the shared genetics were seen for OA and cardiac arrhythmias, OA and CHD, and OA and stroke, whereas unique environmental factors played a larger role for co-occurrence of OA and heart failure." I suggested the authors in the last round to explain more about the finding that unique environmental factors may play a larger role in the common cause of OA and HF. If the authors are not able to comment further on this finding as they mentioned in the response letter that there are very small number HF cases, I therefore suggest the authors tune down a little bit regarding the roles of unique environmental factors on co-occurrence of OA and heart failure.	We agree we could tune down our findings regarding OA and HF because of the small number of HF cases.	We have revised our summary in the beginning of the Discussion section: "Among the included specific CVDs, the shared genetics were seen for OA and cardiac arrhythmias, OA and CHD, and OA and stroke, whereas unique environmental factors may play a larger role for co-occurrence of OA and heart failure. However, for the latter, findings are more uncertain due to few cases."